# The Association between Anthropometric Measurements and Body Composition with Hand Grip Strength among the Elderly Population in Indonesia

**DOI:** 10.3390/jcm13164697

**Published:** 2024-08-10

**Authors:** Nina Kemala Sari, Stepvia Stepvia, Muhana Fawwazy Ilyas

**Affiliations:** 1Geriatric Division, Department of Internal Medicine, Dr. Cipto Mangunkusumo National Referral Hospital, Faculty of Medicine, Universitas Indonesia, Jakarta 10430, Indonesia; stepvia@student.uns.ac.id; 2Faculty of Medicine, Universitas Sebelas Maret, Surakarta 57126, Indonesia; muhanailyas@gmail.com

**Keywords:** anthropometry, calf circumference, fat free mass, hand grip strength, muscle mass, sarcopenia

## Abstract

**Background/Objectives**: Hand grip strength (HGS) is a crucial measure for evaluating muscle function and general physical ability, and it may be associated with several diseases. Previous studies have demonstrated inconsistent associations between anthropometric measurement and body composition with HGS. This study aims to investigate the association between anthropometric measurement and body composition with HGS in the elderly population residing in Indonesia. **Methods**: This is a cross-sectional study on older adults aged between 60 and 82 years who live in the community. Anthropometric parameters assessed in this study comprised the body mass index (BMI), mid-upper arm circumference (MUAC), calf circumference (CC), and waist circumference (WC). Subsequently, body composition measurements, including fat mass (FM), fat-free mass (FFM), muscle mass (MM), skeletal muscle mass (SMM), and the appendicular skeletal mass index (ASMI), were assessed using a body composition analyzer. Last, the measurement of HGS was conducted using a hand dynamometer. **Results**: A total of 109 participants were involved in this study. Our study demonstrates a significant association between anthropometric parameters, namely CC and HGS. Subsequently, several body composition parameters, including FFM, SMM, ASMI, and MM in the four extremities, are also significantly associated with HGS. However, in a multivariate analysis, only CC and FFM were able to significantly predict HGS. **Conclusions**: Improving CC and maintaining FFM may enhance muscle strength in older adults. This suggests that targeted exercise and nutrition programs could increase muscle mass and strength, thereby mitigating age-related decline and improving quality of life.

## 1. Introduction

The aging population is increasing, and the aging process is associated with various health conditions. According to the World Health Organization (WHO), the global elderly population is expected to double by 2050, reaching nearly 400 million people aged 80 and older [1]. The phenomenon of aging is correlated with a decline in physical vigor [2]. Muscular strength is vital in determining physical performance and self-sufficiency among elderly individuals [3]. Previous studies indicate that a decrease in muscle strength is associated with an increased occurrence of cardiovascular [4], metabolic [5], cerebrovascular issues [6], falls [7], early hospitalization [8], and premature death [9].

Various options exist for evaluating muscle strength in elderly individuals, encompassing a range of equipment and methods that test voluntary movements associated with strength [10]. Hand grip strength (HGS) is a widely used and practical measure of the maximal force generated by the hand during a voluntary contraction. It is routinely used to evaluate muscular function [10,11]. Studies have shown a connection between a weak HGS and several negative health consequences, including reduced physical abilities, poor cognitive function, chronic illnesses, and early death [12]. An HGS test provides a valuable clinical assessment tool for evaluating total muscle health. It is essential because it can predict outcomes, is non-invasive, and is reliable and convenient in measuring muscle mass and strength [13]. An HGS test is also user-friendly, efficient, and may be beneficial, especially when utilized in low- and middle-income countries, including Indonesia.

Previous studies have investigated the relation between HGS and various factors. HGS measurement is frequently affected by several factors, including body weight [12,14,15], body height [14,15,16,17], body mass index [14,18], fat mass, fat-free mass [16], waist circumference, and calf circumference [15]. Subsequently, HGS has demonstrated a correlation with body composition characteristics and physical fitness performance [19]. However, studies have shown different results, and the number of studies is still limited, especially in relation to body composition. Furthermore, this study aims to investigate the association between anthropometric measurements and body composition with HGS in the elderly population living in Indonesia. The study hypothesizes significant associations between certain anthropometric measurements and body composition parameters with HGS among the elderly population in Indonesia.

## 2. Materials and Methods

### 2.1. Participants

Participants were older adults between 60 and 82 years of age who lived in the community. A total of 153 individuals were initially selected using purposive sampling based on the eligibility criteria. The inclusion criteria included elderly individuals with no recorded upper limb pathology, no hand, wrist, or forearm discomfort, and no history of neurological disorders affecting the upper quadrant. The exclusion criteria involved those who did not give informed consent, had sensory impairments, finger or hand amputations, active arthritis, hemiplegia, quadriplegia, recent upper extremity surgery within the past three months, or had an acute infection or related symptoms. Finally, 109 subject participants were included in the study.

### 2.2. Study Design

This cross-sectional study was conducted at Cempaka Putih Community Health Center and Johar Baru Community Health Center in Jakarta, Indonesia, from July to August 2023. The study gathered socio-demographic information, pre-existing medical conditions, anthropometric measurements, body composition, and HGS measurements. The Ethics Committee of the Faculty of Medicine, University of Indonesia—Dr. Cipto Mangunkusumo National Referral Hospital gave ethical approval to carry out the study (Ethical Application Ref: 23-11-1936 and date of approval: 15 March 2023).

### 2.3. Measurements

#### 2.3.1. Anthropometric

The anthropometric parameters assessed in this study comprised the body mass index (BMI), mid-upper arm circumference (MUAC), calf circumference (CC), and waist circumference (WC). First, the participants’ height and body mass were measured on two occasions using standardized methods with a precision of 0.1 cm and 0.1 kg, respectively. Subsequently, the BMI was calculated as body mass (kg)/height (m)^2^. The participants were then classified into four groups according to their BMI. Individuals with a BMI of less than 18.5 (kg/m^2^) were labeled as underweight, individuals with a BMI between 18.5 and 24.99 (kg/m^2^) were defined as the normoweight group, and those with a BMI between 25 and 29.99 (kg/m^2^) and equal or more than 30.00 (kg/m^2^) were classified as overweight and obese, respectively. Subsequently, anthropometric tape was utilized to precisely measure the MUAC, CC, and WC with an accuracy of 0.1 cm. The measurements were conducted when the subjects were wearing lightweight garments. The MUAC was measured at the halfway point between the bony prominence of the ulna and the bony prominence of the scapula. The maximal lateral distance around the left calf was measured to quantify the CC while the participant maintained an upright posture. Last, the WC was measured at the midpoint between the lowest rib and the iliac crest on a horizontal plane [20,21]. The investigators (NKS and SS) conducted the assessments in July 2023 during the morning.

#### 2.3.2. Body Composition

The body composition was assessed using a body composition analyzer (Model MC980, modified Tanita Corp., Tokyo, Japan) that utilizes bioelectrical impedance analysis (BIA). An alternating current is used to test the impedance of the human body. Touch resistance occurs when the human body makes touch with the electrode. The task was carried out by operators who have undergone professional training. The device was calibrated 30 min before being used each day. The participants excreted urine during this time, and water consumption was banned in the 30 min leading up to the measurement. We followed all recommended standards, including conducting the BIA assessments in the morning after an overnight fast to ensure accurate and reliable measurements. They were clad in a single garment, with no shoes or socks, and positioned themselves accurately. The participants were directed to securely grasp the handle and place their thumbs on the oval electrode. In addition, they were given directions to maintain their arms in an extended position and refrain from touching any other parts of their bodies. The entire treatment lasted approximately 1 min. The study assessed the quantities of fat mass (FM), fat-free mass (FFM), muscle mass (MM), and skeletal muscle mass (SMM). In addition, the appendicular skeletal mass index (ASMI) was calculated by dividing the appendicular skeletal mass (ASM) by the square of the height (m^2^) [22].

#### 2.3.3. Hand Grip Strength

A JAMAR^®^ hydraulic hand dynamometer (Model J00105, Lafayette Instrument Company, Lafayette, United States of America (USA)) was used to measure the HGS of the dominant hand. The professional operators evaluated the HGS of the participants while they were sitting down, with their elbows bent at a 90-degree angle and their wrists in a neutral posture. Subsequently, aid was provided below the dynamometer. In this role, the person must apply maximum physical strength to compress the HGS dynamometer. HGS quantification entails the measurement of the intensity of the stationary force applied by the hand while compressing or squeezing a dynamometer. The maximum voluntary contraction was sustained for a minimum of 3 s and quantified as the HGS in kg. Three measurements were taken at 60 s intervals, and the average measurement, as well as the highest value, was chosen for analysis [23]. The practice of averaging multiple measurements was performed to ensure the accuracy and reliability of measurement.

### 2.4. Statistical Analysis

First, univariate analysis was performed. The quantitative variables are expressed as mean and standard deviation (SD) values and their normality distribution was measured using the Kolmogorov–Smirnov test. The categorical variables are presented using frequency and percentages. Subsequently, bivariate analysis was performed using an analysis of variance (ANOVA) test to investigate the association between BMI classification and HGS. Pearson’s correlation coefficients were employed to evaluate the associations among anthropometric measurements, body compositions, and HGS. Subsequently, multiple linear regression models were tested to identify variables significantly predicting hand grip strength. Multiple linear regression was performed using a backward approach, resulting in only significant variables included in the last model. The statistical analysis was performed with the software SPSS, Version 25.0 (Armonk, NY, USA: IBM Corp). The level of significance was set at α = 0.05.

## 3. Results

### 3.1. Participant Characteristics

This study included 109 subjects, comprising 85 females and 24 males, aged 66 ± 5.3 years. Out of all participants, only three were diagnosed with sarcopenia. All of the participant characteristics are presented in Table 1. Study participants had a BMI of 25.1 ± 3.7 kg/m^2^, with 45% of the subjects categorized as overweight and 42.2% categorized as normoweight. Subsequently, the MUAC, CC, and WC were 27.1 ± 4.1 cm, 33.1 ± 3.4 cm, and 95.0 ± 12.5 cm, respectively. Regarding body composition, FM, FFM, MM, and SMM were 20.8 ± 8.1 kg, 37.1 ± 6.1 kg, 35.3 ± 5.8 kg, and 20.1 ± 3.6 kg, respectively. In addition, the muscle mass of the four limbs, consisting of the left arm, right arm, left leg, and right leg, were 1.7 ± 0.4, 1.8 ± 0.4, 6.3 ± 1.3, and 6.3 ± 1.3, respectively. Entire anthropometric measurements, including the BMI, MUAC, CC, and WC; body composition, including FM, FFM, MM, SMM, ASMI, and ASM; as well as HGS, including mean HGS and max HGS, are presented in Table 2.

### 3.2. Association between Anthropometric Measurement and Body Composition with Hand Grip Strength

There are positive associations between CC, FFM, SMM, ASMI, and MM in all four limbs with HGS (Table 3). On the other hand, there are no associations in other parameters, including the BMI, MUAC, WC, and FM. Subsequently, there are no significant differences in HGS in each BMI classification (Table 4). Last, in a multivariate analysis, only CC and FFM are able to significantly predict HGS (Table 5).

## 4. Discussion

HGS refers to the quantification of the amount of force exerted by the fingers and thumb while grasping an object. Grip strength can be quantified by the amalgamation of an individual’s capacity to apply force and their ability to maintain it over a period of time [24,25]. HGS is a crucial measure for evaluating muscle function and general physical ability. In addition, it may also be associated with several other illnesses, such as chronic anemia, abnormal lipid levels, metabolic syndrome, chronic kidney disease, hypertension, and other cardiovascular disorders [26,27]. Multiple longitudinal cohort studies conducted in the United States and China have demonstrated a strong correlation between decreased handgrip strength and a higher risk of developing type 2 diabetes [28]. Therefore, studying HGS and its association with various parameters, including commonly assessed and easily obtainable data such as anthropometric measurements and body composition, is crucial for understanding and improving health outcomes in the elderly population.

The present investigation revealed no significant difference in HGS between various BMI classifications. Subsequently, there was no association between BMI and HGS, which shows results that differ from those of other studies. A study conducted in India [29] found different results and revealed that individuals of both sexes with a healthy body weight demonstrated higher hand muscle strength. The possible cause is dietary conditions that directly impact both the strength and quality of muscles. The decrease in muscular strength among persons with a low BMI can also be attributed to insufficient consumption of calories. Subsequently, in individuals who are overweight or obese, the decline in muscle strength is likely caused by the infiltration of body fat into muscle tissues and alterations in the organization of type I and type II muscle fibers, ultimately affecting muscle strength [29]. Inadequate muscular strength is frequently observed in individuals with low body weight and suboptimal dietary health [30]. Another investigation also revealed that an individual’s nutritional condition impacts their body weight, which is closely linked to the strength of their handgrip [31].

Multiple studies have also found a correlation between HGS and BMI in different genders and across age groups. However, some other studies have not identified any such link [32]. This study encompassed individuals spanning diverse age groups, genders, nationalities, jobs, and nutritional preferences. Research has demonstrated that HGS is not just determined by BMI but is also influenced by gender, age, and height [33]. Based on another study [34], it has been found that overweight individuals may have a higher proportion of muscle mass compared to fat. Additional examination of the muscle biopsy and the state of the muscle fibers could provide useful insights into this matter, considering the perplexing findings of various research. The BMI, as an index, does not have the capacity to distinguish between changes in weight that are caused by changes in muscle mass or body fat percentage [35,36].

The current investigation revealed associations between FFM, SMM, ASMI, and MM in all four extremities with HGS. Steffl et al. [37] conducted a study to investigate the correlation between HGS, MM, and physical performance in a group of 69 older women living in the community. The findings demonstrated a strong correlation between the variables and determined that MM and HGS were the only significant predictors of sarcopenia compared to other physical fitness statuses. The connection between these factors can be elucidated by the release of adipocytokines from adipose tissue, which can induce persistent inflammation in both systemic and localized regions of the body. This inflammation may be associated with decreased functional capacity and a decline in muscle strength [38]. HGS quantifies the level of muscular strength in the individual muscle fibers. Furthermore, augmented muscular hypertrophy leads to enhanced force production, leading to an elevation in HGS. On the other hand, excessive fat buildup can result in a reduction in MM, leading to a subsequent decline in HGS. In addition, the accumulation of adipose tissue within the muscles has been identified as a contributing factor to reduced muscular strength [39]. Last, there was proof that CC can significantly predict HGS. Another study has cast doubt on the relationship between CC and physical performance. An analysis of 572 consecutive patients hospitalized at geriatric outpatient clinics in Amsterdam found a correlation between CC and physical function tests, including HGS [40]. On the other hand, the findings of our study were consistent with the research conducted by Wang et al. [41]. Therefore, based on the findings of this research group, it is suggested that CC may also be a potentially useful screening method for individuals who are more susceptible to developing sarcopenia.

Despite the valuable insights provided by this study, several limitations must be acknowledged. First, the cross-sectional design precludes the establishment of causal relationships between anthropometric measurements, body composition, and HGS. Future studies should aim to include a more balanced gender representation and larger sample sizes to enhance the robustness and applicability of the results [42]. Another limitation pertains to the measurement tools and protocols used in the study. While BIA is a convenient and non-invasive method for assessing body composition, it is less accurate than more sophisticated techniques such as dual-energy X-ray absorptiometry or Computed tomography scan. Ensuring rigorous standardization of measurement protocols and incorporating more precise body composition assessment methods in future studies would also enhance the validity and reliability of the findings.

## 5. Conclusions

This study highlights the significant associations between certain anthropometric measurements (CC) and body composition parameters (FFM, SMM, ASMI, and MM in all four extremities) with HGS in the elderly population of Indonesia. Specifically, CC and FFM emerged as significant predictors of HGS, underscoring the importance of maintaining muscle mass and appropriate body composition for preserving muscle function and overall physical ability in older adults. These findings also suggest that simple anthropometric measurements, such as CC, could serve as potential screening tools for identifying individuals at risk of reduced muscle strength and related functional decline.

These findings suggest that simple anthropometric measurements such as CC could serve as practical screening tools for identifying individuals at risk of reduced muscle strength and related functional decline. Interventions aimed at improving muscle mass and optimizing body composition, such as resistance training and nutritional support, may enhance physical function and reduce the risk of frailty and associated comorbidities in this population. Future studies should continue to explore these associations in diverse elderly cohorts and investigate the effectiveness of targeted interventions in improving HGS and overall health outcomes in older adults.

## Figures and Tables

**Table 1 jcm-13-04697-t001:** Baseline characteristics of study participants.

Characteristics	Total Participants (*n =* 109)
Ages (years)	66.2 ± 5.3
Sarcopenia (*n*)	3 (2.8%)
Rheumatology disease (*n*)	21 (19.3%)
Hypertension (*n*)	31 (28.4%)
Coronary heart disease (*n*)	1 (0.9%)
Diabetes mellitus (*n*)	18 (16.5%)
Chronic obstructive pulmonary disease (*n*)	2 (1.8%)
Gastrointestinal disease (*n*)	8 (7.3%)
Liver disease (*n*)	2 (1.8%)
Hypercholesterolemia (*n*)	10 (9.2%)
Hyperuricemia (*n*)	6 (5.5%)
Cerebrovascular disease (*n*)	3 (2.7%)
Glaucoma (*n*)	1 (0.9%)
Vertigo (*n*)	2 (1.8%)
Hernia nucleus pulposus (*n*)	2 (1.8%)
Cancer (*n*)	1 (0.9%)

Data are presented as mean ± SD for continuous variables or frequency and percentage for categoric variables.

**Table 2 jcm-13-04697-t002:** Anthropometric measurement, body composition, and hand grip strength of study participants.

Variables	Measurement	Total Participants (*n* = 109)
Anthropometric Measurement	Body weight (kg)	58.1 ± 9.9
	Height (cm)	152.0 ± 7.4
	Body mass index (kg/m^2^)	25.1 ± 3.7
	Underweight (*n*)	4 (3.7%)
	Normoweight (*n*)	46 (42.2%)
	Overweight (*n*)	49 (45.0%)
	Obesity (*n*)	10 (9.2%)
	Mid-upper arm circumference (cm)	27.1 ± 4.1
	Calf circumference (cm)	33.1 ± 3.4
	Waist circumference (cm)	95.0 ± 12.5
Body Composition	Fat mass (kg)	20.8 ± 8.1
	Fat-free mass (kg)	37.1 ± 6.1
	Muscle mass (kg)	35.3 ± 5.8
	Skeletal muscle mass (kg)	20.1 ± 3.6
	Left arm muscle mass (kg)	1.7 ± 0.4
	Right arm muscle mass (kg)	1.8 ± 0.4
	Left leg muscle mass (kg)	6.3 ± 1.3
	Right leg muscle mass (kg)	6.3 ± 1.3
	Appendicular skeletal muscle index (kg/m^2^)	6.9 ± 0.9
Hand Grip Strength	Mean value (kg)	19.7 ± 6.4
	Maximal value (kg)	21.1 ± 6.7

Data are presented as mean ± SD for continuous variables or frequency and percentage for categoric variables.

**Table 3 jcm-13-04697-t003:** Correlation between anthropometric measurement and body composition with hand grip strength.

Variables		Mean HGS		Maximum HGS	
		r	*p* *	r	*p* *
Anthropometric Measurement	Body mass index (kg/m^2^)	−0.02	0.83	−0.02	0.83
	Mid-upper arm circumference (cm)	0.08	0.43	0.97	0.32
	Calf circumference (cm)	0.18	0.04 *	0.19	0.04 *
	Waist circumference (cm)	0.03	0.74	0.20	0.84
Body Composition	Fat mass (kg)	−0.17	0.08	−0.17	0.08
	Fat-free mass (kg)	0.65	0.00 *	0.66	0.00 *
	Muscle mass (kg)	0.61	0.00 *	0.63	0.00 *
	Skeletal muscle mass (kg)	0.61	0.00 *	0.62	0.00 *
	Left arm muscle mass (kg)	0.58	0.00 *	0.58	0.00 *
	Right arm muscle mass (kg)	0.54	0.00 *	0.55	0.00 *
	Left leg muscle mass (kg)	0.55	0.00 *	0.56	0.00 *
	Right leg muscle mass (kg)	0.58	0.00 *	0.59	0.00 *
	Appendicular skeletal muscle index (kg/m^2^)	0.46	0.00 *	0.46	0.00 *

HGS, hand grip strength. The association between variables was analyzed using the Pearson correlation test. * *p*-value < 0.05 indicates statistical significance.

**Table 4 jcm-13-04697-t004:** Comparison of hand grip strength between different BMI classifications.

Classification		Mean HGS			Max HGS		
		Mean ± SD	F Value	*p*-Value	Mean ± SD	F Value	*p*-Value
Total (*n =* 109)	Underweight (*n =* 4)	12.7 ± 2.7	2.27	0.08	13.5 ± 1.9	2.42	0.07
	Normoweight (*n =* 46)	20.8 ± 6.8			22.3 ± 7.3		
	Overweight (*n =* 49)	19.2 ± 6.1			20.7 ± 6.5		
	Obesity (*n =* 10)	19.3 ± 4.7			20.5 ± 4.0		
Female (*n =* 85)	Underweight (*n =* 4)	12.7 ± 2.7	2.07	0.11	13.5± 1.9	2.33	0.08
	Normoweight (*n =* 29)	17.6 ± 3.9			18.8 ± 3.9		
	Overweight (*n =* 43)	18.1 ± 4.6			19.4 ± 5.0		
	Obesity (*n =* 9)	18.7 ± 4.5			20.0 ± 3.8		
Male (*n =* 24)	Underweight (*n =* 0)		0.09	0.91		0.13	0.88
	Normoweight (*n =* 17)	26.3 ± 7.3			28.4 ± 7.8		
	Overweight (*n =* 6)	27.7 ± 9.1			29.3 ± 9.2		
	Obesity (*n =* 1)	25.0 ± 0.0			25.0 ± 0.0		

HGS, hand grip strength; SD, standard deviation. The differences between groups were analyzed using the one-way ANOVA test.

**Table 5 jcm-13-04697-t005:** Multiple linear regression for hand grip strength.

Dependent Variable	Independent Variables	β	SE	*p*-Value	95% CI for B
Mean HGS	Calf circumference	−0.32	0.16	0.05 *	−0.63–−0.01
	Fat-free mass	0.76	0.09	0.01 *	0.59–0.93
Maximum HGS	Calf circumference	−0.32	0.16	0.05 *	−0.64–0.01
	Fat-free mass	0.80	0.09	0.01 *	0.63–0.98

CI, confidence interval; HGS, hand grip strength; SE, standard error. Multiple linear regression was performed using a backward approach; only the last model is visualized in the table. * *p*-value < 0.05 indicates statistical significance.

## Data Availability

The data supporting this study’s findings are available from the corresponding author upon reasonable request.

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
