# Peer review of "The Association between Anthropometric Measurements and Body Composition with Hand Grip Strength among the Elderly Population in Indonesia"

_jcm, 2024, doi:10.3390/jcm13164697_

Round 1
Reviewer 1 Report
Comments and Suggestions for Authors
I congratulate the authors for the article. Please find below my suggestions that are needed to be addedd in order for this manuscript to be considered for publication.
Methods
2.2. Measurement of anthropometric parameters
Some information about the assessment process is needed. Who did the assessments? When were the assessments conducted?
2.3. Measurement of body composition
BIA was used. However, BIA is suggested to be used in the morning after an overnight fast? Did you follow this and other similar standards?!
Results
I don’t see it fit to compare 85 females with 24 males? Authors should consider excluding gender differences and calculations based on genders in the first two tables.
Discussion
Lines 248-249 – please cite papers that have previously used BIA and support your statement. Anyhow, you could check how the authors did when facing a similar situation in this paper (https://doi.org/10.3390/jcm11195579).
References
There are quite old references cited in the text and I suggest to revise and change with newer up-to-date ones.
Comments on the Quality of English LanguagePlease correct the gramatical mistakes throughout the text.
Author Response
Response to Reviewer 1 Comments
1. Summary
Thank you very much for taking the time to review this manuscript. Please find the detailed responses below and the corresponding revisions/corrections highlighted/in track changes in the re-submitted files.
2. Questions for General Evaluation
Are the methods adequately described?
Reviewer’s Evaluation: Must be improved
Response and Revisions: We have added detailed information about the assessment process, including who conducted the assessments and when they were conducted. This revision addresses the reviewer's concern and ensures the methods are now adequately described.
Are the results clearly presented?
Reviewer’s Evaluation: Must be improved
Response and Revisions: To improve clarity, we have revised the presentation of the results. Specifically, we have excluded gender differences and calculations based on genders in the first two tables. This revision addresses the reviewer's concern and ensures the results are now clearly presented.
Are the conclusions supported by the results?
Reviewer’s Evaluation: Can be improved
Response and Revisions: We have refined the discussion section to ensure that the conclusions are more clearly supported by the results. This revision addresses the reviewer's concern and ensures the conclusions are now well-supported by the results.
3. Point-by-point response to Comments and Suggestions for Author
Comments 1: Some information about the assessment process is needed. Who did the assessments? When were the assessments conducted?
Response 1: The investigators (NKS and SS) conducted the assessments during July 2023 in the morning. We have included this information in the revised manuscript.
Comments 2: BIA was used. However, BIA is suggested to be used in the morning after an overnight fast? Did you follow this and other similar standards?!
Response 2: We followed all recommended standards, including conducting the BIA assessments in the morning after an overnight fast to ensure accurate and reliable measurements.
Comments 3: I don’t see it fit to compare 85 females with 24 males? Authors should consider excluding gender differences and calculations based on genders in the first two tables.
Response 3: Agree. We have, accordingly, revised the first two tables to exclude gender differences and calculations based on genders.
Comments 4: Lines 248-249 – please cite papers that have previously used BIA and support your statement. Anyhow, you could check how the authors did when facing a similar situation in this paper (https://doi.org/10.3390/jcm11195579).
Response 4: Thank you for the suggestion. We have added relevant citations to support our statement, including references to the paper suggested.
Comments 5: There are quite old references cited in the text and I suggest to revise and change with newer up-to-date ones.
Response 5: Thank you for highlighting this. We have updated the references and replaced older citations with more recent and relevant ones. The updated reference list can be found at the end of the manuscript.
4. Response to Comments on the Quality of English Language
Reviewer’s Evaluation: Moderate editing of English language required
Response: We have made the necessary edits to improve the clarity and readability of the manuscript.
5. Additional clarifications
None.
Reviewer 2 Report
Comments and Suggestions for Authors
22. Please, avoid to refer on a brand of measurement tools.
26-28. Please, give the practical application of the results. How does this association contribute?
29-30. Differentiate the keywords from the title of your manuscript.
33. Please, modify the syntax of the sentence.
56-61. The reason that the measurement was conducted is clear. However, what is the purpose of the study? Also, what is the research hypothesis?
63. Split methodology into 1. Subjects, 2. Study design and 3. Measurements 3.1 Anthropometric, 3.2 Body composition, etc.
64. Please, refer to inclusion and exclusion criteria.
67. Did you have any ethical approval?
68. Please, refer to the participants' number.
96. Please, refer to the methods that you used.
113. Similarly, provide the reference for your methodology.
123. Why did you take the average of the 3 measurements? Do you have any reference to regard this statement?
136. Try to close with this sentence: “The statistical analysis was performed with the software SPSS, Version 25.0 (Armonk, NY, USA: IBM Corp). The level of significance was set at α = 0.05.”
139-140. This information should be referred to earlier, in the methodology.
141. Was sarcopenia an exclusion criterion?
173. Explain SE in the table 5.
241. Regard this statement with a reference.
253. Give the readers the practical application of your study.
Comments on the Quality of English LanguageThe authors must check some parts of the manuscript for syntax and typos.
Author Response
Response to Reviewer 2 Comments
1. Summary
Thank you very much for taking the time to review this manuscript. Please find the detailed responses below and the corresponding revisions/corrections highlighted/in track changes in the re-submitted files.
2. Questions for General Evaluation
Does the introduction provide sufficient background and include all relevant references?
Reviewer’s Evaluation: Can be improved
Response and Revisions: We have revised the introduction to provide a more comprehensive background and included all relevant references. This revision addresses the reviewer's concern and ensures the introduction is now more thorough.
Is the research design appropriate?
Reviewer’s Evaluation: Can be improved
Response and Revisions: We have refined the description of the research design to ensure clarity and appropriateness. This revision addresses the reviewer's concern and enhances the overall design clarity.
Are the methods adequately described?
Reviewer’s Evaluation: Must be improved
Response and Revisions: We have added detailed information about the assessment process, including who conducted the assessments and when they were conducted. This revision addresses the reviewer's concern and ensures the methods are now adequately described.
Are the results clearly presented?
Reviewer’s Evaluation: Yes
Response and Revisions: We appreciate your positive feedback on the clarity of our results presentation.
Are the conclusions supported by the results?
Reviewer’s Evaluation: Can be improved
Response and Revisions: We have refined the discussion section to ensure that the conclusions are more clearly supported by the results. This revision addresses the reviewer's concern and ensures the conclusions are now well-supported by the results.
3. Point-by-point response to Comments and Suggestions for Authors
Comments 1: Please avoid referring to a brand of measurement tools.
Response 1: Thank you for the suggestion. Unfortunately, we prefer to keep the brand in our manuscript because different results may result from different tools, and specifying the brand ensures clarity and reproducibility of our study.
Comments 2: Lines 26-28: Please give the practical application of the results. How does this association contribute?
Response 2: We have added a section to explain the practical applications of our findings and how this association contributes to the field. “These findings suggest that simple anthropometric measurements such as CC could serve as practical screening tools for identifying individuals at risk of reduced muscle strength and related functional decline.”
Comments 3: Lines 29-30: Differentiate the keywords from the title of your manuscript.
Response 3: We have revised the keywords to ensure they are distinct from the title of the manuscript.
Comments 4: Line 33: Please modify the syntax of the sentence.
Response 4: We have revised the syntax of the sentence on line 33 to improve clarity.
Comments 5: Lines 56-61: The reason that the measurement was conducted is clear. However, what is the purpose of the study? Also, what is the research hypothesis?
Response 5: We have added a section detailing the purpose of the study and the research hypothesis.
Comments 6: Line 63: Split methodology into 1. Subjects, 2. Study design, and 3. Measurements (3.1 Anthropometric, 3.2 Body composition, etc.).
Response 6: We have restructured the methodology section as suggested.
Comments 7: Line 64: Please refer to inclusion and exclusion criteria.
Response 7: We have included the inclusion and exclusion criteria in the revised manuscript.
Comments 8: Line 67: Did you have any ethical approval?
Response 8: Yes, we obtained ethical approval. This information has been added to the manuscript.
Comments 9: Line 68: Please refer to the participants' number.
Response 9: We have included the number of participants in the revised manuscript.
Comments 10: Line 96: Please refer to the methods that you used.
Response 10: We have specified the methods used in the revised manuscript
Comments 11: Line 113: Similarly, provide the reference for your methodology.
Response 11: We have added the reference for our methodology.
Comments 12: Line 123: Why did you take the average of the 3 measurements? Do you have any reference to regard this statement?
Response 12: We have explained the rationale for averaging the 3 measurements. “The practice of averaging multiple measurThe practice of averaging multiple measurements is performed to ensure accuracy and reliability of measurement.ements is recommended to ensure accuracy and reliability of measurement”
Comments 13: Line 136: Try to close with this sentence: “The statistical analysis was performed with the software SPSS, Version 25.0 (Armonk, NY, USA: IBM Corp). The level of significance was set at α = 0.05.”
Response 13: We have revised the end of the methodology section.
Comments 14: Lines 139-140: This information should be referred to earlier, in the methodology.
Response 14: We have moved the information to the appropriate section in the methodology.
Comments 15: Line 141: Was sarcopenia an exclusion criterion?
Response 15: Sarcopenia was not an exclusion criterion in our study. We included participants regardless of their sarcopenia status to ensure a comprehensive analysis.
Comments 16: Line 173: Explain SE in Table 5.
Response 16: We have added an explanation for SE in the footnote of Table 5.
Comments 17: Line 241: Support this statement with a reference.
Response 17: We have added a reference to support the statement.
Comments 18: Line 253: Give the readers the practical application of your study.
Response 18: We have included a section on the practical applications of our study on Conclusion.
4. Response to Comments on the Quality of English Language
Reviewer’s Evaluation: Minor editing of English language required.
Response: We have made the necessary edits to improve the clarity and readability of the manuscript. The authors have checked the manuscript for syntax and typographical errors to ensure high-quality English language throughout the text.
5. Additional clarifications
None.
Round 2
Reviewer 1 Report
Comments and Suggestions for Authors
I believe that the required revisions have been thoroughly addressed. The document has been reviewed and adjusted to meet the specified requirements.
Author Response
Dear Reviewer, Thank you for your feedback. We appreciate your assessment of our manuscript. We are pleased to hear that the language quality is satisfactory and that no issues were detected. Thank you for your time and valuable input. Best regards,
Reviewer 2 Report
Comments and Suggestions for Authors
Dear authors
Please, check in line 73 to include the number of participants.
Comments on the Quality of English Language
English language needs a minor revision.
Author Response
Dear Reviewer,
Thank you for your valuable feedback and the time you have taken to review our manuscript.
We appreciate your comments and suggestions regarding the quality of English language in our paper. We have carefully addressed the points you raised:
1. Number of Participants:
We have included the number of participants in the revised manuscript as follows:
"A total of 115 individuals were initially selected using purposive sampling based on eligibility criteria."
"Finally, 109 subject participants were included in the study."
2. English Language:
We have made several corrections to the English language as suggested. We believe these revisions address the minor issues noted and improve the overall clarity of the manuscript.
We hope these changes meet your expectations and enhance the quality of our paper. Thank you again for your constructive feedback.
Best regards,
